

# Leveraging JEDI for Atmospheric Composition: A unified framework for evaluating observations and model forecasts

Shih-Wei Wei[1,2], Jérôme Barré[3,4], Soyoung Ha[5], Cheng-Hsuan Lu[1,2], Maryam Abdi-Oskouei[3,4], Benjamin Ménétrier[6], Cheng Dang[1]

[1]Joint Center for Satellite Data Assimilation, University Corporation for Atmospheric Research, Boulder Colorado, USA
[2]University at Albany, Albany, New York, USA
[3]Global Modeling and Assimilation Office, NASA Goddard Space Flight Center, Greenbelt, Maryland, USA
[4]Goddard Earth Sciences Technology and Research II, Morgan State University, Baltimore, Maryland, USA
[5]Mesoscale and Microscale Meteorology Laboratory, National Center for Atmospheric Research, Boulder, Colorado, USA
[6]Norwegian Meteorological Institute, Oslo, Norway

*Correspondence to*: Soyoung Ha (syha@ucar.edu) and Shih-Wei Wei (swei@ucar.edu)

**Abstract.** Accurate evaluation of both observations and forecasts is essential for advancing atmospheric composition research and improving operational prediction. Traditionally, this has relied on separate workflows with product-specific preprocessing, often limiting reproducibility and creating inconsistencies between models and observational datasets or across different products. Modern data assimilation systems provide precise observation operators for mapping model variables into observation space, yet these capabilities remain underutilized outside assimilation. Here, we demonstrate how the Joint Effort for Data assimilation Integration (JEDI) framework addresses this gap by offering a unified, modular, and model-agnostic system that integrates data assimilation with systematic evaluation. JEDI enables consistent intercomparisons of observations, forecasts, and reanalyses by interfacing with diverse forecast models and gridded datasets, while leveraging carefully designed observation operators to compute equivalent quantities for a wide range of observation products. These include satellite instruments such as Tropospheric Emissions: Monitoring of Pollution (TEMPO), The TROPOspheric Monitoring Instrument (TROPOMI), Moderate Resolution Imaging Spectroradiometer (MODIS), Visible Infrared Imaging Radiometer Suite (VIIRS), and Plankton, Aerosol, Cloud, and ocean Ecosystem (PACE), as well as ground-based networks like Aerosol Robotic Network (AERONET), Pandora, and U.S. Environmental Protection Agency (EPA) AirNow. Case studies illustrate the flexibility of this workflow: (1) $NO_2$ forecasts from the Weather Research and Forecasting model coupled with Chemistry (WRF-Chem) evaluated against TEMPO, TROPOMI, and Pandora retrievals; (2) surface fine particulate matter and ozone forecasts from WRF-Chem assessed against AirNow measurements using EPA regulatory thresholds; and (3) aerosol optical depth (AOD) retrievals from multiple satellites compared with Modern-Era Retrospective analysis for Research and Applications, Version 2 (MERRA-2) and validated against AERONET. These examples highlight JEDI's ability to detect systematic regional biases, reconcile complementary sampling characteristics across platforms, and assess the added value of unified observation operators for cross-comparison. Overall, JEDI provides a consistent and extensible framework for model validation and observation assessment, reducing redundant preprocessing and





aligning evaluation with operational data assimilation practices, and ultimately advancing both research and operational applications in atmospheric composition.

## 1 Introduction


With the growth of Earth system observations from both spaceborne and ground-based platforms, new types of products enabled by advanced instruments and retrieval algorithms now provide increasingly detailed snapshots of atmospheric composition and dynamics. However, these observations are inherently limited: they represent discrete samples in space and time, they do not provide continuous global coverage, and they are restricted to the present and past. In addition,

observations are subject to random and systematic errors arising from instrument precision, calibration uncertainties, and sensor degradation over time. As a result, observations alone are insufficient for comprehensive characterization of the Earth system state. Numerical models remain indispensable for filling these gaps. They produce three-dimensional fields at regular intervals, offering spatially and temporally continuous representations of processes across the Earth system in the past, present, and future—capabilities that are fundamentally beyond observational datasets. These include numerical weather

prediction (NWP) models extended with atmospheric composition, global chemistry–climate models, and chemical transport models. Such models have been widely used for both operational forecasting and research. Yet, as with observations, model representations of the earth system are prone to errors as they remain constrained by underlying assumptions, approximations, and parameterizations that limit their accuracy and predictive skill.

In this context, it is essential to make systematic comparisons between observations and models. Such evaluation not only

identifies systematic errors and uncertainties but also strengthens confidence in the use of datasets for prediction and analysis (Levy et al. 2013; Giles et al., 2019). Evaluation and data assimilation (DA) are intrinsically linked: observations are used to update (or initialize) the model state for forecasts, while forecasts can serve as both a baseline for assessing those observations or as a priori for retrieval algorithms (Bocquet et al., 2010).

Traditionally, evaluation has been performed outside of DA frameworks, often relying on ad hoc approaches or stand-alone

verification tools, in which observations are compared against model simulations or, conversely, models are evaluated against observational datasets. Depending on the application, the baseline may be a single model used to assess multiple observational products, or a set of observations used to intercompare multiple models. For example, the MELODIES-MONET system (Baker and Pan, 2017) and the METplus system (Jensen et al., 2024) provide flexible platforms for model–observation comparison and verification, but they generally require product-specific preprocessing and rely sometimes on

simplified transformations that are not fully representing the observation characteristics and or the underlying model physics. In contrast, observation operators developed within DA systems are explicitly designed to represent the observational characteristics and physical processes that link model state variables to observed quantities. Because errors in these operators propagate during the DA procedure, they require the highest precision and consistency (Courtier et al., 1998; Bannister, 2017). Leveraging them for evaluation therefore provides a unique opportunity to evaluate observations and models within a



unified DA framework. Without such integration in evaluation frameworks, separate forward models must be built, requiring redundant efforts and introducing additional sources of uncertainty. In that sense leveraging DA capabilities for evaluation purposes with well-vetted observation operators ensures consistency and reliability across both applications (Kalnay 2003; Carrassi et al. 2018).

The Joint Effort for Data assimilation Integration (JEDI; Trémolet and Auligné, 2020) framework provides exactly this
capability. Designed as a flexible, modular, and model-agnostic system, JEDI integrates observation operators, standardized data formats, and statistical tools to enable seamless comparison of forecasts, analyses, and diverse observational datasets through common interfaces. Its modular design allows users to engage with specific components without needing to master the full system. Although originally developed for DA, JEDI also supports consistent processing and evaluation of diverse observational datasets independent of assimilation. Built on object-oriented and generic programming principles, it facilitates
side-by-side comparisons and cross-platform integration, creating new opportunities for unified evaluation and intercomparison studies.

In this paper, we showcase the evaluation capability of JEDI across a range of atmospheric composition datasets. Case studies demonstrate comparisons between forecast models, reanalysis fields, and diverse observational products, including satellite retrievals, ground-based networks, and regulatory air quality monitoring, with minimal code development. The
examples presented here highlight both the diagnostic power of JEDI's observation operators and its broader potential as a common system for evaluation, verification, and cross-comparison in atmospheric composition research and operations. The paper is organized as follows: Section 2 describes the evaluation framework and its components; Section 3 and 4 introduce the observation and model datasets used for this demonstration; Section 5 presents use cases for various atmospheric composition applications such as regional air quality and global aerosol composition using $NO_2$ satellite retrievals, surface
pollutants, and aerosol optical depth (AOD) observations; finally Section 6 provides a summary.

## 2 Evaluation framework description

### 2.1 A model agnostic system

The system described in this paper is mainly based on the JEDI (Trémolet and Auligné, 2020; Abdi-Oskouei and Barré, 2025) maintained by the Joint Center for Satellite Data Assimilation (JCSDA). The JEDI is a modular framework designed
to support diverse DA and evaluation applications. Built with object-oriented and generic programming techniques, JEDI consists of independent components, allowing users to interact with specific DA aspects without needing to master in depth the entire system. Central to JEDI is the Object-Oriented Prediction System (OOPS), which provides abstraction for the DA building blocks (e.g., forecast models and observations with their respective errors) to interact modularly. By separating algorithm design from model- or observation-specific details, OOPS allows integration of different numerical models and
observation datasets, while supporting variational, ensemble, and hybrid assimilation approaches. With that extended capability, the current system description leverages parts of the entire JEDI framework.



The overall workflow is illustrated in Figure 1. Observations from satellite granules or ground stations are first converted into the common Interface for Observation Data Access (IODA) format using Python-based converters. IODA files store metadata (e.g., latitude, longitude, time, level), observation values, observation errors, quality control (QC) flags, and ancillary data (averaging kernels, scattering weights, a priori) in a standardized structure. Model forecasts states (conventionally denoted x) are ingested through the gridded data interface VIND (Versatile Implementation for Native Data), which allows spatial interpolation onto the gridded state at observation locations to provide Geophysical Variables at Locations (GeoVaLs), i.e. the full model vertical profiles for a set of required variables. The Unified Forward Operator (UFO) then applies the observation operator (conventionally denoted H) on the GeoVaLs to derive model equivalents, H(x), which are stored in new IODA files alongside the original observations. The differences between observations and H(x) provide the basis for assimilation and also evaluation.

For evaluation, verification and statistics, resulting H(x) IODA files can be directly processed by users' own diagnostic and visualization tools or passed to a verification package, such as METplus (Jensen et al., 2024) via a Python utility that converts the paired data into METplus-readable format. Additional diagnostic and visualization tools are needed to process the statistical outputs produced by METplus.



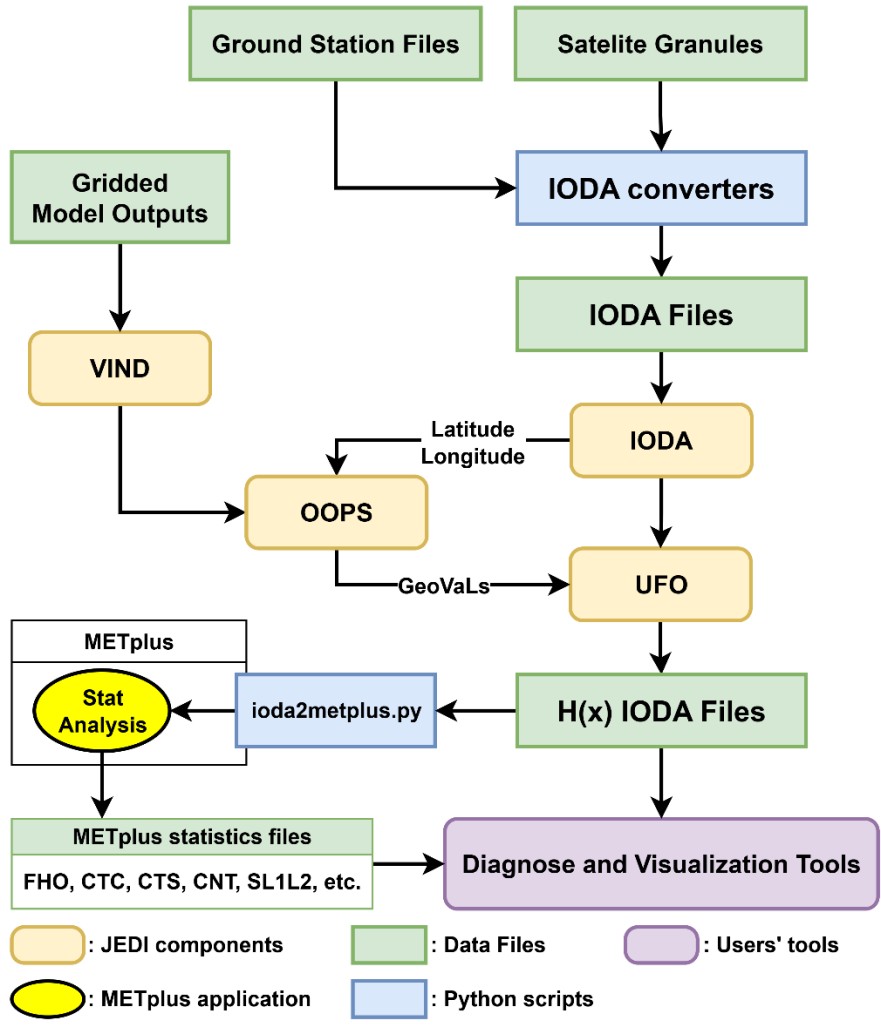

**Figure 1. Flowchart of the evaluation system combining JEDI and METplus (FHO for Forecast, Hit, Observation Rates; CTC for Contingency Table Counts; CTS for Contingency Table Statistics; CNT for Continuous Statistics; SL1L2 for Scalar L1L2 Partial Sums).**

## 2.2 IODA and UFO

The IODA provides a unified data format within JEDI for storing and exchanging observational datasets. IODA files organize information into standardized groups, including MetaData (e.g., latitude, longitude, time, pressure level), ObsValue (measured quantities), and optional groups such as quality flags or error estimates. This consistent structure allows diverse observations (from satellite retrievals to in situ measurements) to be handled by the same tools and workflows. To generate these files, IODA converters are used to transform native observation formats (e.g., NASA satellite granules, EPA AirNow CSVs, or campaign data) into IODA-compliant files. These converters are typically implemented in Python and maintained



within the JCSDA repositories (see code and data availability section), ensuring reproducibility and compatibility across applications. By adopting the IODA convention, JEDI facilitates interoperability, reduces the need for custom preprocessing scripts, and enables a seamless connection between observational data and forward operators.

The UFO plays a key role in linking model states to observations by simulating model equivalent quantities to observations. This capability enables direct comparison of forecasts with actual observations during DA cycles. The UFO offers a consistent, extensible interface that supports a wide variety of observation types, ranging from satellite radiances processed through radiative transfer models to satellite retrieval products to in situ measurements. Beyond forward modeling, the UFO also handles key pre- and post-processing tasks such as filtering, quality control, bias correction, and the computation of 
observation-minus-background (observations relative to the model state) and observation-minus-analysis (observations relative to the completed data assimilation procedure) quantities.

The UFO is designed to operate independently of any specific forecast model interface. As mentioned in Section 2.1, it receives GeoVaLs as input, which are background state variables interpolated in space and time to the observation points by the model interface (see Section 2.3). Model interfaces within JEDI are responsible for producing GeoVaLs, while OOPS 
coordinates the transfer of information from the model to UFO. Once provided, the observation operator can perform further vertical processing, such as interpolation, coordinate transformations, or integration, depending on the nature of the measurements. This architecture keeps the interpolation process uniform across models while allowing each observation operator to encapsulate its own specialized logic, promoting flexibility, maintainability, and scalability. The following sections describe the three observation operators applied in this study.

### 2.2.1 Identity and Vertical interpolation


The identity operator in UFO is used to generate the H(x) quantities at the surface with only horizontal interpolation involved during the forward calculation. This operator as default directly extracts the lowest model level from the concentrations GeoVaLs. If users wish to perform a comparison with measurements over altitudes the vertical interpolation is also available in UFO, this has not been used in this paper but widely used in other JEDI applications (i.e., Abdi-Oskouei 
and Barré, 2025). For additional technical details are provided in the JEDI online documentation:

https://jointcenterforsatellitedataassimilation-jedi-docs.readthedocs-hosted.com/en/latest/inside/jedi-components/ufo/obsops.html#obsops-identity   and   https://jointcenterforsatellitedataassimilation-jedi-docs.readthedocs-hosted.com/en/latest/inside/jedi-components/ufo/obsops.html#vertical-interpolation

### 2.2.2 Column retrieval

The column retrieval operator in UFO provides a generalized capability for assimilating vertically integrated atmospheric retrievals from satellite, airborne, and ground-based sensors. By utilizing averaging kernels and a priori profiles included with retrieval products, the operator maps model state variables, such as trace gas mixing ratios, into the observation space, producing partial or total column quantities. This design supports a consistent two-step assimilation methodology across



multiple platforms. In this work, we apply the column retrieval operator to various $NO_2$ retrievals (see Section 3),
demonstrating its ability to handle various observational datasets. Additional technical details are provided in the JEDI
online documentation: https://jointcenterforsatellitedataassimilation-jedi-docs.readthedocs-hosted.com/en/latest/inside/jedi-
components/ufo/obsops.html#column-retrieval-operator

### 2.2.3 CRTM AOD

The Community Radiative Transfer Model (CRTM) AOD operator in JEDI provides the capability to simulate satellite-
retrieved aerosol quantities directly from model aerosol mixing ratios. Within UFO, CRTM is applied in a simplified
configuration using pre-defined lookup tables (LUTs). These LUTs specify mass extinction coefficients for each aerosol
species over a range of wavelengths, which are used as relative weights for mass concentration of each species to compute
total aerosol AOD at each vertical level. With this design CRTM AOD allows for consistent forward modeling of diverse
aerosol products, accounting for species-specific optical properties under assumptions of particle shape, size distributions,
and refractive indices. In this work, we used the Goddard Chemistry Aerosol Radiation and Transport (GOCART) GEOS5
LUT with CRTM. While currently demonstrated application relies on the GEOS-5 LUTs to approximate aerosol
microphysics, the operator provides a flexible framework for future enhancements, including the use of other operational
aerosol models or coupling with online microphysics schemes. Additional technical information of this operator can be
found at: https://jointcenterforsatellitedataassimilation-jedi-docs.readthedocs-hosted.com/en/latest/inside/jedi-
components/ufo/obsops.html#aerosol-optical-depth-aodcrtm

### 2.3 Versatile Implementation for Native Data (VIND)

While observations are processed through IODA and UFO components, as described above, numerical forecast models
require a different set of common interfaces within JEDI. Regardless of the model variables and native coordinates used in
each model, they must implement OOPS abstract interfaces for shared components (for example what defines the model
geometry, the state x, the model forecast and so on) in order to use the generic algorithms available in JEDI. To this end, the
Versatile Implementation for Native Data (VIND) provides all required interfaces for shared components except for the
forecast step. As a result, VIND can run any OOPS-based DA method that does not require a prediction model or its tangent-
linear/adjoint formulation (e.g., in 4D-Var). VIND builds on ATLAS, an open-source C++ library from ECMWF
(Deconinck et al., 2017; https://sites.ecmwf.int/docs/atlas/), which provides data structures for handling fields on a wide
range of global and regional grid geometries. Integrating a new modeling system into VIND requires little to no additional
development, apart from implementing lightweight file readers and writers for model field data if existing ones are not
already compatible. This simplicity, together with broad grid compatibility provided by ATLAS, makes VIND an attractive
entry point for new users of the JEDI framework.



### 2.4 METplus

The Model Evaluation Tools (MET; Brown et al., 2021; Prestopnik et al., 2025) is a community-supported software package developed by the Developmental Testbed Center (DTC) to provide standardized methods for verifying NWP forecasts. MET includes a suite of applications for traditional grid-to-grid and grid-to-observation comparisons, statistical diagnostics, and visualization of forecast skill. Building on MET, METplus (Jensen et al., 2024) is a Python-based wrapper that streamlines the configuration and execution of MET tools through a modular, workflow-oriented interface. By abstracting the underlying

MET applications into reusable components, METplus reduces the technical burden on users and enables flexible chaining of tasks for complex evaluation pipelines.

Within the atmospheric composition context, METplus allows users to compute a wide range of statistics—such as bias, root-mean-square error, contingency table counts, and categorical verification scores—using paired model and observational data. Its flexible configuration system supports both deterministic and ensemble forecasts and can be adapted to regulatory

thresholds (e.g., EPA breakpoints for $PM_{2.5}$ and ozone). The role of METplus is primarily in post-processing and verification; it does not provide complex observation operators to generate H(x) over a variety of model outputs. Instead, these functions are supplied through JEDI's IODA data format and observation operators. Together, these components provide a unified evaluation framework in which atmospheric composition forecasts can be systematically compared with in-situ and satellite retrievals, thereby bridging the gap between research-oriented and operational verification practices.

## 3 Observations

### 3.1. TEMPO $NO_2$ tropospheric columns

The Tropospheric Emissions: Monitoring of Pollution (TEMPO) instrument is a geostationary ultraviolet–visible (UV–Vis) spectrometer that provides hourly observations of atmospheric pollutants over North America at $2 \times 4.75$ $km^2$ resolution (Zoogman et al., 2017). Operating from 91°W longitude over the 290–740 nm spectral range, TEMPO measures key trace

gases including nitrogen dioxide ($NO_2$), ozone, and formaldehyde. $NO_2$ retrievals are derived by estimating slant column densities (SCDs), removing the stratospheric contribution using GEOS-CF and DA constraints, and converting to vertical column densities (VCDs) via air mass factors (AMFs) that account for viewing geometry, surface reflectance, and atmospheric profiles (Nowlan et al., 2016). Algorithm details are provided in the TEMPO ATBD (Nowlan et al., 2025). In this study, we use version 3 Level 2 tropospheric $NO_2$, retaining only pixels with quality flag being zero and cloud fraction

less than 0.5 (https://doi.org/10.5067/IS-40e/TEMPO/NO2_L2.003).

### 3.2. TROPOMI $NO_2$ tropospheric columns

The TROPOspheric Monitoring Instrument (TROPOMI), onboard Sentinel-5 Precursor since October 2017, is a nadir-viewing hyperspectral sensor operating in the ultraviolet and visible (UV–Vis) with spatial resolution improved from $3.5 \times 7$



km$^2$ to 3.5 × 5.5 km$^2$ in 2019, enabling urban-scale mapping of NO$_2$ (Eskes et al., 2022). NO$_2$ retrievals use the Differential
Optical Absorption Spectroscopy (DOAS) technique to derive SCDs, which are converted to VCDs via AMFs based on
radiative transfer and a priori profiles (Veefkind et al., 2012). The product includes both tropospheric and stratospheric NO$_2$
and is widely applied in air quality and model validation studies. In this work, we use version 2.4.0 of the Level 2
tropospheric NO$_2$ product, following the User Manual (Eskes et al., 2022) recommendation to retain only pixels with quality
assurance value above 0.75, thereby excluding cloud fractions above 0.5 (https://doi.org/10.5270/S5P-9bnp8q8).

**3.3 PANDORA NO$_2$ total columns**

The Pandora spectrometer system is a ground-based remote sensing instrument designed to provide high-resolution
measurements of trace gases such as NO$_2$ and ozone. Operating in the UV-Vis spectral range, Pandora instruments retrieve
total column abundances by analyzing direct solar irradiance spectra, and can also produce tropospheric columns when
combined with a priori information and stratospheric corrections (Herman et al., 2009). Pandora systems have been deployed
in networks such as the Pandonia Global Network (PGN), which supports long-term monitoring and satellite validation
efforts. In this study, total column NO$_2$ retrievals from Pandora are used to complement satellite observations, offering
higher temporal resolution (with retrievals available at intervals of seconds to minutes) and localized ground-truth
measurements.

**3.4. AirNow ozone and PM$_{2.5}$ at surface**

The AirNow system, operated by the U.S. Environmental Protection Agency (EPA), delivers near-real-time, surface-level air
quality observations across the United States, focusing on key regulatory pollutants such as ozone and fine particulate matter
(PM$_{2.5}$). These data are collected from a network of fixed monitoring stations operated by federal, state, tribal, and local air
quality agencies. Observations are reported hourly and undergo preliminary quality control to ensure reliability for
operational applications and scientific analysis.

**3.5. OCI AOD**

The Plankton, Aerosol, Cloud, and ocean Ecosystem (PACE) mission was successfully launched in February 2024 (Werdell
et al., 2019). PACE provides a broad suite of datasets through its primary sensor, the Ocean Color Instrument (OCI). In this
paper we use AOD retrievals from OCI using the Unified Aerosol Algorithm (UAA; Remer et al. 2019a, b). The UAA
inherits Dark Target (DT; Levy et al., 2024) and Deep Blue (DB; Hsu et al., 2013) algorithms for the ocean and land surface,
respectively, and provides AOD retrievals at 0.354, 0.388, 0.48, 0.55, 0.67, 0.87, 1.24, 1.64, and 2.2 μm. Version 3
(https://doi.org/10.5067/PACE/OCI/L2/AER_UAA/3.0) of the product is used here, while version 3.1 was released during
the preparation of this draft. The released versions of the UAA AOD product remain in testing mode and are not yet
recommended for scientific applications. Once validated, however, these datasets will offer diverse spatial, spectral, and
biogeophysical insight across Earth's ocean–atmosphere–land interface, providing unique opportunities to characterize global



aerosol properties from space. In addition to AOD, various other observation types (https://pace.oceansciences.org/data_table.htm#23) can be readily incorporated into JEDI with minimal implementation effort. By enabling retrieval assessment and unified intercomparison, the JEDI framework can play a pivotal role in maximizing the value of these novel datasets for both observational studies and forecasting applications.

### 3.6. MODIS AOD

The Level 2 AOD retrievals from Moderate Resolution Imaging Spectroradiometer (MODIS) Collection 6.1 (C6.1; Levy and Hsu et al., 2015) on satellite Terra (MOD04_L2) and Aqua (MYD04_L2) are processed in this study. It provides the combined product for AOD at 550 nm from Dark Target (Levy et al., 2013) and Deep Blue (Hsu et al., 2013) algorithms, which were originally developed for different surface types. When the Normalized Difference Vegetation Index (NDVI) is larger than 0.3, DT retrievals are provided. When NDVI is smaller than 0.2, DB retrievals are provided. For the remaining
pixels, the average of DT and DB retrievals or the available one passing the recommended quality assurance, which is 3 for DT and 2 for DB, is used. The retrieval is based on 20 by 20 pixels at the blue band (500 m resolution), resulting in a resolution of 10 km at nadir.

### 3.7. VIIRS AOD

The Visible Infrared Imaging Radiometer Suite (VIIRS), onboard the Suomi-NPP and NOAA-20 satellites, provides Level 2
aerosol optical depth (AOD) retrievals using two complementary algorithms: DT (Levy et al., 2015; Sawyer et al., 2020) and DB (Hsu et al., 2013; Lee et al., 2024). The DT algorithm retrieves AOD over dark, vegetated land surfaces and oceans by exploiting the low surface reflectance in the visible and shortwave infrared bands, while the DB algorithm extends AOD retrievals to bright-reflecting surfaces such as deserts and arid regions, where DT is less reliable. Together, these products deliver near-global AOD coverage at a native resolution of approximately 6 km, enabling detailed monitoring of aerosol
distributions. In this study, we utilize both DT and DB AOD products from Suomi-NPP and NOAA-20 to evaluate aerosol simulations, benefiting from their complementary spatial coverage and surface-type sensitivity.

### 3.8. AERONET AOD

The Aerosol Robotic Network (AERONET) provides high-quality ground-based observations of AOD through a global network of sun photometers. In this study, we use Level 1.5 AOD measurements from AERONET version 3 (Giles et al.,
2019), which include cloud-screened and quality-assured retrievals at multiple wavelengths. AERONET data serve as an essential reference for evaluating aerosol products and observation operators, offering well-calibrated, long-term records with high temporal resolution.



## 4 Gridded inputs

### 4.1 WRF-Chem

A week of hourly forecasts from WRF-Chem v4.5.2 (Grell et al., 2005; Skamarock et al., 2021) system, focused on New York State and the surrounding areas, is used to demonstrate the case for trace gases evaluation. The system uses the National Center for Environmental Prediction (NCEP) Global Forecast System (GFS) forecasts as its meteorological initial and lateral boundary conditions. To constrain the meteorological conditions, a fully cycle 3-dimensional variational DA system based on the Gridpoint Statistical Interpolation (GSI) (Kleist et al., 2009) is implemented. It assimilates conventional

data in the NCEP Global Data Assimilation System (GDAS) and the New York State Mesonet (NYSM) surface meteorological data (Brotzge et al., 2020) and wind profiler observations (Shrestha et al., 2021) every 6 hours at 00, 06, 12, and 18 UTC.

For chemical initial and lateral boundary conditions, 6-hourly 0.9º x 1.25º forecasts from the Whole Atmosphere Community Climate Model's (WACCM; Gettleman et al., 2019) are used. Biogenic emissions are calculated by the Model of Emissions

of Gases and Aerosols from Nature (MEGAN; Guenther et al., 2006), and biomass burning emissions are supplied from the FIre Inventory from NCAR (FINN) version 2.5.1 (Wiedinmyer et al., 2023). Anthropogenic emissions are based on the EPA National Emission Inventory (NEI) 2016 modeling platform. The chemistry scheme used in the system is T1-MOZCART, which couples the MOZART-T1 (Emmons et al., 2020) with GOCART aerosols (Chin et al., 2003; Colarco et al., 2010).

### 4.2 MERRA-2

MERRA-2 (Gelaro et al., 2017; Randles et al., 2017) is the reanalysis dataset produced by the NASA Global Modeling and Assimilation Office (GMAO) based on the GEOS-5 system. It assimilates AOD measurements from AERONET, the Multiangle Imaging Spectro Radiometer (MISR), MODIS, and the Advanced Very-High-Resolution Radiometer (AVHRR) instruments. Note that the MODIS AOD product assimilated in MERRA-2 is bias corrected via a neural network algorithm, which is a different product from the MODIS C6.1 used in this study. MERRA-2 data is publicly available at NASA's

Goddard Earth Sciences and Information Services Center (https://gmao.gsfc.nasa.gov/gmao-products/merra-2/data-access_merra-2/). In this study, preprocessing is performed to combine 3-hourly reanalysis of meteorological conditions (https://doi.org/10.5067/WWQSXQ8IVFW8) and aerosol mixing ratios (https://doi.org/10.5067/LTVB4GPCOTK2).

## 5 Use Cases

### 5.1 Space and ground-based NO₂ retrievals vs WRF-Chem

Figure 2 compares tropospheric NO₂ retrievals from TEMPO and TROPOMI with an example of a 1-hour window centered at 19 UTC 24 August 2024 against corresponding WRF-Chem forecasts. This example shows the potential for evaluating forecast performance using satellite data, which is quantified by innovation statistics, i.e., the differences between simulated

observation equivalents, H(x), and retrievals. In this example, TEMPO's higher spatial resolution provides finer detail, while discrepancies between TEMPO and TROPOMI are evident over Boston, western Pennsylvania, and southern New Jersey

areas. WRF-Chem underestimates NO$_2$ tropospheric columns over the New York metropolitan area, overestimates near Montreal and Vermont, and shows better agreement with elevated concentrations near Toronto. This highlights the capability to cross-compare different satellite products and model outputs.

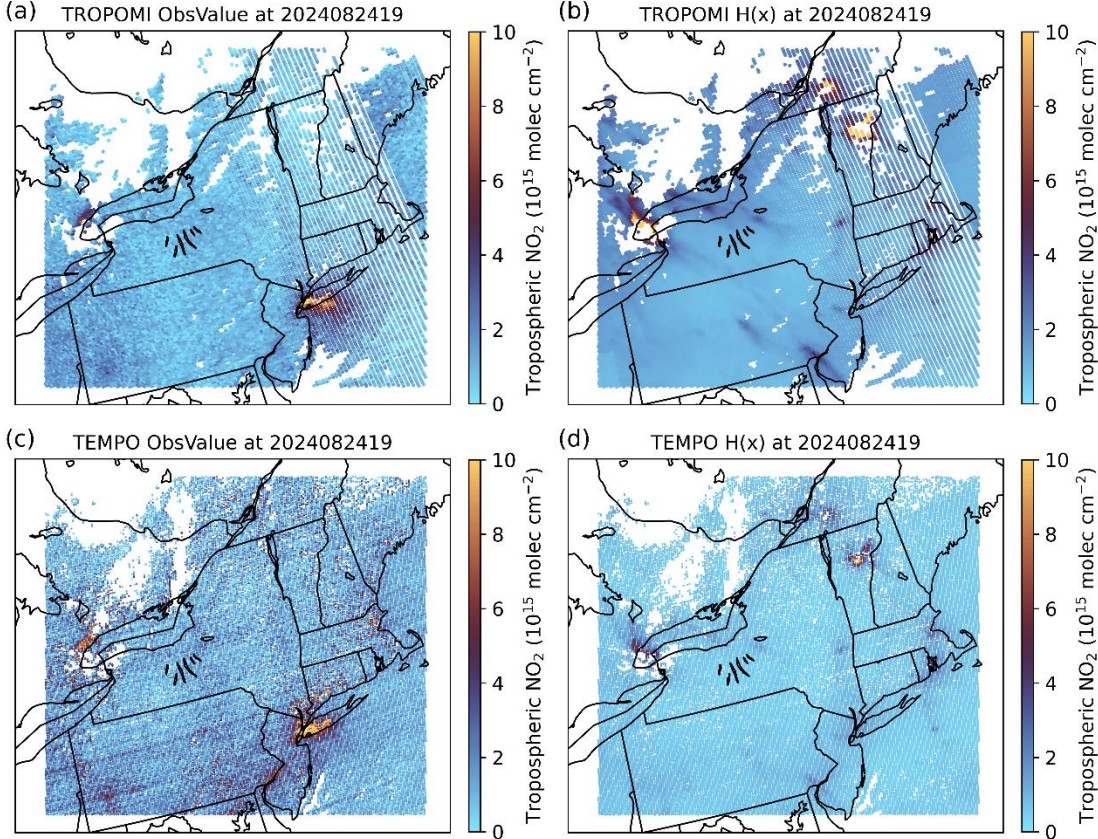

**Figure 2. Spatial distribution of tropospheric NO$_2$ retrievals from (a) TROPOMI ObsValue and (b) TROPOMI H(x) and (c)**
**TEMPO ObsValue and (d) TEMPO H(x) valid between 13:30 and 14:30 LST (18:30 and 19:30 UTC) 24 August 2024. H(x) is based on the 19-hour forecast valid at 14:00 LST (19:00 UTC) 24 August 2024 from WRF-Chem.**

Bias and root-mean-square error (RMSE) statistics are computed over time in Figure 3. This offers insight into systematic over- or underestimation patterns in time that can inform on atmospheric model process and emission inventory improvements. The temporal sampling of TEMPO and TROPOMI provides complementary perspectives. TEMPO, in

geostationary orbit, delivers hourly coverage across North America, capturing diurnal cycles and short-lived events. In contrast, polar low earth orbiting TROPOMI provides only one revisit per day around 1:30 PM local time for NO$_2$ measurements, therefore is incapable of providing insights on diurnal variability.



**Figure 3. Time series of (a) Bias and (b) RMSE derived from hourly differences of tropospheric NO₂ between retrievals from satellite sensors (blue: TROPOMI; orange: TEMPO) and H(x) from WRF-Chem from 22 August to 1 September 2024. Mean values are provided in parentheses.**

Beyond NO₂ columns, the JEDI column retrieval operator supports evaluation of other trace gases. Tested cases include total and partial columns of carbon monoxide (CO) and ozone, demonstrating its flexibility across species with varying averaging kernels and a priori information dependencies. Although not shown here, these applications highlight the broader utility of





JEDI for multi-species analysis. Looking ahead, the same framework can be extended to retrievals of formaldehyde (HCHO) and sulfur dioxide ($SO_2$), key for studying photochemistry and emission sources. While not yet tested, their integration is expected to be straightforward within JEDI's modular design.

Figure 4 shows the PANDORA total column $NO_2$ retrievals and corresponding H(x) based on WRF-Chem forecasts. The temporal resolution of PANDORA retrievals can vary from a few seconds to a couple of minutes, which is higher than the

hourly outputs from WRF-Chem. WRF-Chem did not capture the high $NO_2$ concentration event over New York City and its surrounding areas, while it shows better agreement with PANDORA retrievals in Boston and the west coast of Lake Ontario. In this use case, we produced H(x) within a 1-hour window centered at 19Z 24 August 2024. The type of H(x) demonstrated in this study follows what is typically used in a 3DVar assimilation. All observations in a given time window or more commonly called assimilation window time are compared to a single time background or model file. In our case study the

observations are available every 6 seconds but we have hourly model output files. We average the results comparisons over the hourly time window. Note that if more frequent model output is available, users can adjust the window length (e.g., 30 minutes) for the VIND application.

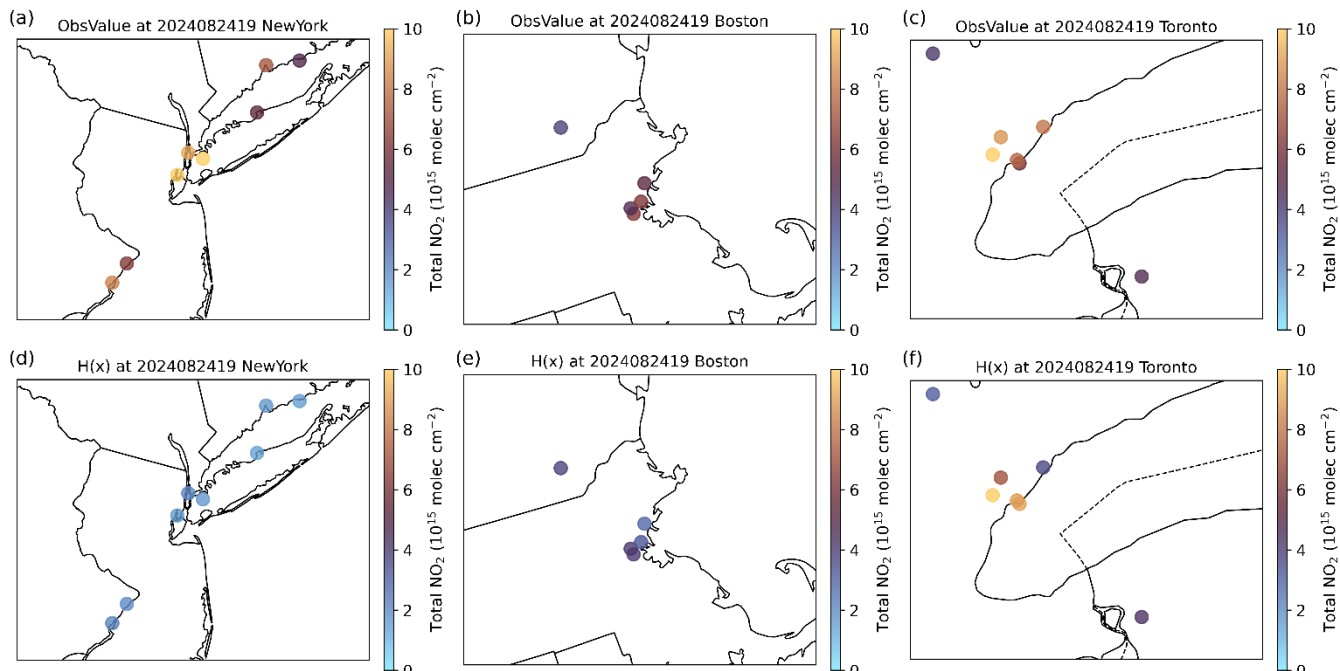

**Figure 4. Spatial distribution of PANDORA total column $NO_2$ retrieval observations and H(x) over (a, d) New York, (b, e) Boston,**
**and (c, f) Toronto areas valid between 13:30 and 14:30 LST (18:30 and 19:30 UTC) 24 August 2024. H(x) is based on the 19-hour forecast valid at 14:00 LST (19:00 UTC 24) August 2024 from WRF-Chem.**

**5.2 AirNow (PM$_{2.5}$ and ozone) vs WRF-Chem**

To evaluate and monitor surface levels of pollution, model forecasts and surface observations are typically evaluated against U.S. EPA regulatory thresholds. In this study, hourly forecasts from WRF-Chem were compared with hourly EPA AirNow





measurements of PM$_{2.5}$ and ozone. Although EPA air quality standards are defined for 24-hour average PM$_{2.5}$ and 8-hour running mean ozone, we applied these breakpoints directly to hourly output data, enabling a consistent contingency table framework for categorical evaluation.

To facilitate this analysis, we developed a Python interface to the StatAnalysis tool in METplus (https://metplus.readthedocs.io/en/latest/Users_Guide/wrappers.html#statanalysis) that reads paired model–observation data

in IODA format. This workflow leverages the existing statistical capabilities of METplus, such as the computation of categorical counts based on user-defined thresholds, while maintaining compatibility with the JEDI-based H(x) outputs. Users can specify thresholds corresponding to regulatory breakpoints or other criteria of interest, allowing flexible evaluation of model skill across multiple categories. Using the observation locations shown in Figure 5 and the same WRF-Chem model outputs as in Section 5.1, an example of resulting contingency statistics is provided in Table 1.

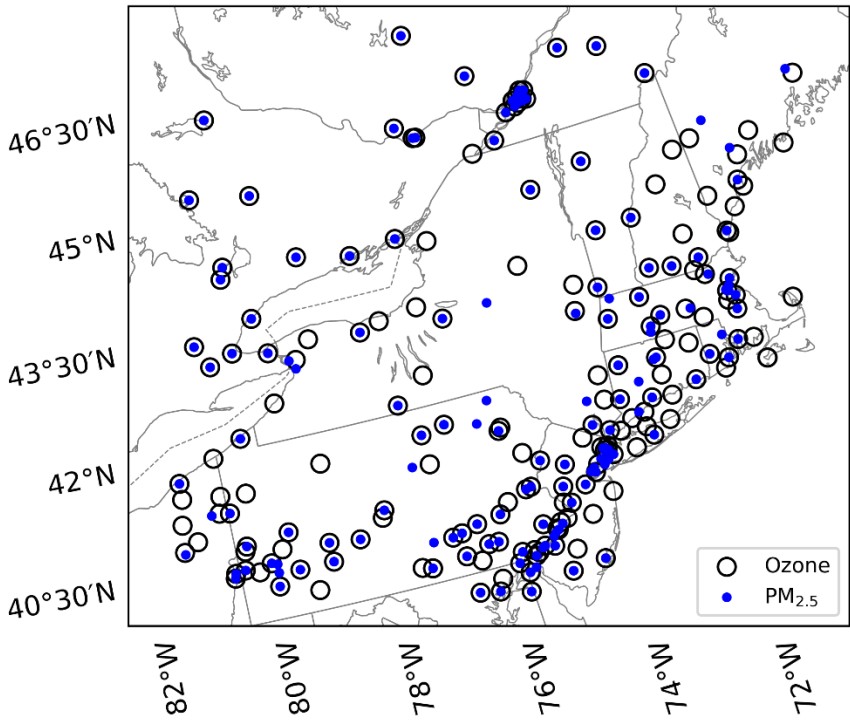


**Figure 5. Available AirNow sites within the domain of the WRF-Chem system from 22 August to 1 September 2024. Sites with ozone measurements in black circles; PM$_{2.5}$ in blue dots.**

Here we showcase how JEDI's UFO observation operators, IODA observation format, and METplus' statistical analysis tools can be combined into a unified approach to assess forecast performance against regulatory air quality standards. This

integration supports operational-style evaluations of air quality models, linking process-based research with metrics directly relevant to public health and policy.

| PM$_{2.5}$ (µg/m$^3$) | Good (0 - 9) | Moderate (9 - 35.4) | Unhealthy (> 35.4) |
| --- | --- | --- | --- |





| Counts = 43273 | O_Y | O_N | O_Y | O_N | O_Y | O_N |
|---|---|---|---|---|---|---|
| F_Y | 19096 (44.13%) | 16057 (37.11%) | 1041 (2.41%) | 7073 (16.35%) | 0 (0%) | 6 (0.01%) |
| F_N | 7009 (16.20%) | 1111 (2.57%) | 15979 (36.93%) | 19180 (44.32%) | 28 (0.06%) | 43239 (99.92%) |
| Ozone (ppb) | Good (0 - 54) | | Moderate (54 - 70) | | Unhealthy (> 70) | |
| Counts = 52488 | O_Y | O_N | O_Y | O_N | O_Y | O_N |
| F_Y | 0 (0%) | 0 (0%) | 0 (0%) | 0 (0%) | 415 (0.79%) | 52073 (99.21%) |
| F_N | 46646 (88.87%) | 5842 (11.13%) | 4228 (8.06%) | 48260 (91.94%) | 0 (0%) | 0 (0%) |

**Table 1. Example of a contingency table of surface PM$_{2.5}$ and ozone based on the EPA breakpoints table. (F: WRF-Chem forecasts, O: AirNow measurements, F/O_Y: Forecast or observation falls into the category, F/O_N: Forecast or observation does not fall into the category).**

### 5.3 AOD products vs. MERRA-2

In this section, we demonstrate the capability of the JEDI system to evaluate AOD retrievals against reanalysis products, using the MERRA-2 reanalysis as a common baseline. The UFO/CRTM AOD operator with GEOS-5 LUTs was applied to aerosol mixing ratios from MERRA-2 to generate model equivalents, H(x), at 550 nm. These were compared with multiple satellite AOD retrievals, including PACE OCI UAA, MODIS C6.1, and VIIRS DT and DB, as well as ground-based AERONET observations.

Figure 6 illustrates an example of the spatial distribution of observation-minus-background average (1-30 November 2024), with MERRA-2 reanalysis as the background, binned onto MERRA-2 grids (0.5° × 0.625°). Overall, AOD H(x) values from MERRA-2 is systematically lower than all the retrievals, with notable underestimation over India, Brazil, and tropical Africa. Regional patterns also emerge, with biases evident over North America, the Sahara Desert, and Australia, as well as fire-active regions such as central Africa and the Amazonia. These comparisons underscore the value of systematic intercomparisons across retrieval products using a consistent reanalysis baseline.



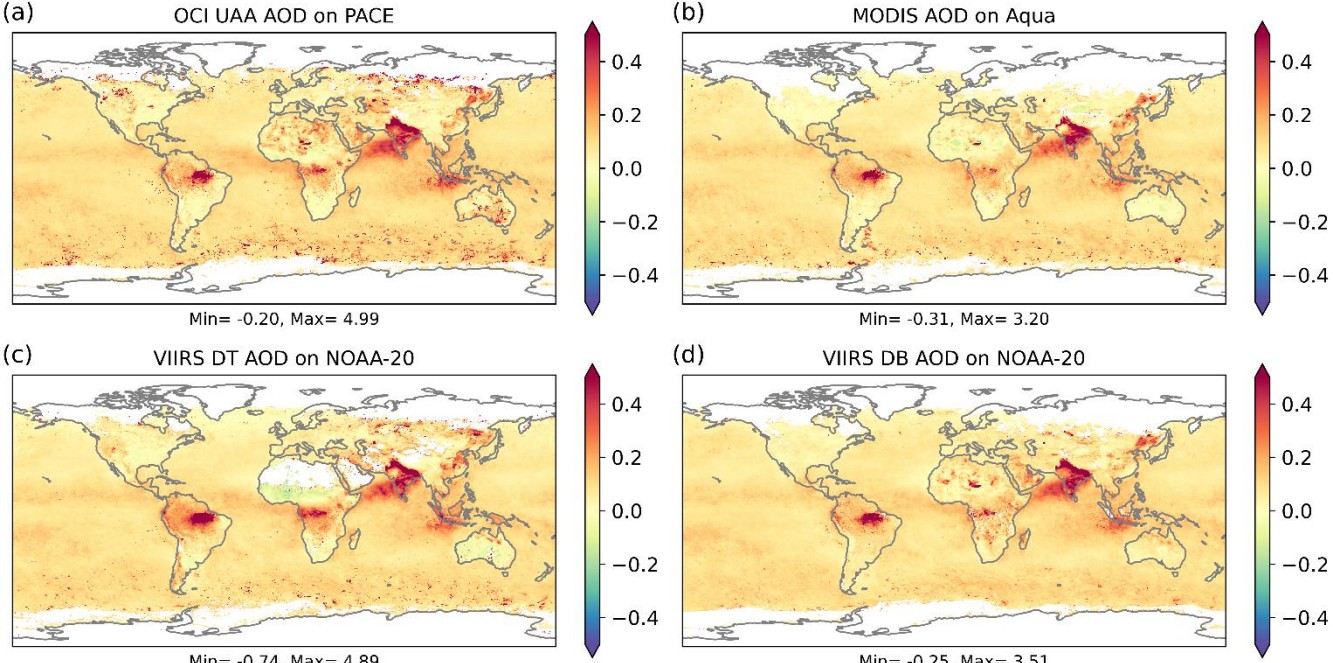

**Figure 6. Averaged observation-minus-background aggregated on MERRA-2 grids (0.5° x 0.625°) for 1-30 November 2024 from (a) OCI UAA AOD on PACE, (b) MODIS AOD on Aqua, (c) VIIRS Dark Target AOD on NOAA-20, and (d) VIIRS Deep Blue**
**AOD on NOAA-20.**

To quantify performance, Figure 7 presents two-dimensional density distributions of AOD observations against H(x) on MERRA-2 reanalyses, against each AOD product. The AOD from MERRA-2 shows better agreement with AERONET than with any of the satellite retrievals, exhibiting higher correlations, lower RMSE, and smaller biases (Table 2). This highlights for example the utility of AERONET as a benchmark for both model evaluation and observation operator performance.
Figure 8 displays the spatial distribution of available AERONET sites and its data volume used for the verification.







**Figure 7. Density distribution plots of AOD H(x) at 550 nm (500 nm for AERONET) on MERRA-2 reanalyses (y-axis) against AOD observation products (x-axis) from (a) OCI UAA on PACE, (b) MODIS C6.1 on Aqua, (c) VIIRS Dark Target on NOAA-20, (d) VIIRS Deep Blue on NOAA-20, and (e) AERONET Level 1.5. Axes are in the logarithmic scale.**



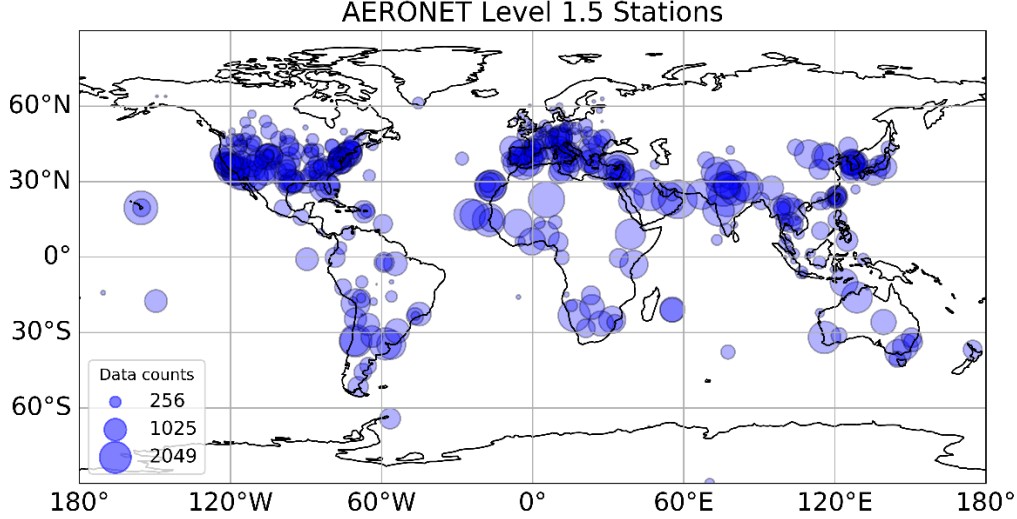


**Figure 8. The site map of AERONET Level 1.5 data for November 2024.**

An additional advantage of the IODA/UFO workflow is that metadata flags, such as land–water classification, are preserved throughout the JEDI procedure and retained in the output files, enabling categorized statistical analysis. This capability allows stratified statistics to be generated without the need for external file matching or pixel collocation. Table 2

summarizes bias, RMSE, and correlation ($R^2$) for each dataset relative to MERRA-2, including separate land–water statistics for the satellite products. These comparisons highlight differences in retrieval performance across surface types and underscore the complementary strengths of each dataset.

|  | **OCI UAA PACE** | **MODIS Aqua** | **MODIS Terra** | **VIIRS DT NOAA-20** |
|---|---|---|---|---|
| **Bias** | -0.093 (-0.088, -0.096) | -0.075 (-0.059, -0.081) | -0.088 (-0.071, -0.095) | -0.070 (-0.070, -0.070) |
| **RMSE** | 0.171 (0.225, 0.132) | 0.137 (0.187, 0.114) | 0.157 (0.203, 0.132) | 0.141 (0.228, 0.102) |
| **$R^2$** | 0.432 (0.423, 0.448) | 0.407 (0.402, 0.421) | 0.386 (0.401, 0.403) | 0.439 (0.421, 0.458) |
|  | **VIIRS DB NOAA-20** | **VIIRS DT S-NPP** | **VIIRS DB S-NPP** | **AERONET Level 1.5** |
| **Bias** | -0.079 (-0.089, -0.074) | -0.101 (-0.110, -0.098) | -0.077 (-0.071, -0.080) | -0.035 |
| **RMSE** | 0.154 (0.214, 0.106) | 0.207 (0.265, 0.187) | 0.157 (0.216, 0.110) | 0.144 |
| **$R^2$** | 0.504 (0.472, 0.513) | 0.312 (0.430, 0.185) | 0.451 (0.427, 0.521) | 0.633 |

**Table 2. Summary table of Bias, RMSE, and $R^2$ of AOD H(x) using MERRA-2 reanalysis against each AOD product. Numbers in parentheses are statistics for land and water, respectively. Separate statistics over land and water are not calculated for**
**AERONET Level 1.5 AOD**

It is important to note that this section does not present a full scientific assessment of AOD products or reanalysis performance. Rather, it demonstrates the comprehensive capability of the JEDI system to integrate diverse aerosol datasets,



generate consistent H(x) simulations using CRTM as the observation operator, and produce diagnostic statistics through a unified IODA framework. This functionality provides a powerful tool for routine evaluation and intercomparison of AOD products in both research and operational contexts.

## 6 Summary

This study demonstrates the capability of the JEDI framework to serve not only as a DA system but also as a unified and flexible platform for the evaluation of atmospheric composition observations and models. Leveraging JEDI's modular design, we used the already existing suite of observation operators to generate observation equivalents H(x) from both WRF-Chem forecasts and the MERRA-2 reanalysis. These H(x) quantities were then compared with a diverse set of ground-based and satellite observation products. The demonstrated comparisons include AirNow, Pandora, TEMPO, TROPOMI, MODIS, VIIRS, PACE, and AERONET and can be extended to much more observations products for not only atmospheric composition but also other application components of the earth system (weather, ocean, land surface).

Through case studies, we illustrated how the system can (1) evaluate WRF-Chem forecasts against high-resolution $NO_2$ retrievals from TEMPO, TROPOMI, and Pandora, (2) assess surface $PM_{2.5}$ and ozone WRF-Chem predictions relative to AirNow monitoring networks using EPA regulatory thresholds, and (3) intercompare multiple satellite AOD products against MERRA-2 reanalysis and AERONET. The examples highlight systematic differences between gridded model outputs and satellite observations, revealing retrieval biases that depend on region and surface type. Such examples also demonstrate the added value of complementary satellite products, such as TEMPO's hourly geostationary coverage versus TROPOMI's global polar-orbiting sampling, in characterizing temporal and spatial variability.

A central strength of this workflow lies in the use of the IODA data format and UFO observation operators, which are designed for data assimilation and therefore aim to perform with the best possible precision. While the UFO provides a model-agnostic framework through JEDI's standardized interfaces, its operators apply appropriate algorithmic and physical transformations and keep modularity and genericity. For instance, in atmospheric composition applications, UFO allows a choice of radiative transfer model and a choice of aerosol optical properties for AOD computations. Also, correct vertical integration and weighting function smoothing that is generic to most gas phase satellite nadir retrieval products is available. Such operators have been designed in a sophisticated and accurate manner to fulfil the modern data assimilation requirements of precision, genericity and built-in quality control filtering functionality. This paper demonstrates how to leverage this already existing capability within JEDI. In combination with METplus statistical tools, JEDI enables efficient generation of standard verification metrics, contingency tables, and stratified statistics based on observation metadata. This avoids the need for product-specific preprocessing and facilitates side-by-side evaluation across multiple instruments and models.

While our focus was on demonstrating system capability rather than delivering a comprehensive scientific assessment, the framework is readily extensible. Future developments include expanding observation conversion to the IODA format for



additional trace gases such as HCHO and $SO_2$, refining aerosol optical property representations in the CRTM AOD operator, and extending the workflow across diverse Earth system models interfaced with VIND and JEDI. By unifying data assimilation infrastructure with systematic evaluation, JEDI provides a robust foundation for advancing atmospheric composition research and enabling more effective use of rapidly growing satellite and ground-based observing systems in scientific and operational contexts. Looking ahead, JEDI also provides a powerful environment for testing the design and

impact of observing systems through Observing System Experiments (OSEs) and Observing System Simulation Experiments (OSSEs). By enabling the generation of synthetic observations and evaluation of their impact, JEDI not only advances the assessment of existing observing systems but also opens new pathways for exploring and shaping future observing strategies.

**Code and data availability**

The JEDI-ACE and related MERRA-2 data processing code has been made available on GitHub

(https://github.com/weiwilliam/JEDI-ACE.git). Users can follow the instruction to check out the commits for VIND and JEDI components. The sample input data and the output of H(x) IODA files and METplus statistics files have been made public available on Zenodo at https://doi.org/10.5281/zenodo.17058099 (Wei et al., 2025). The code for WRF-Chem v4.5.2 has been made publicly available through GitHub (https://github.com/wrf-model/WRF/releases/tag/v4.5.2).

Here we list the products used in this work: TEMPO $NO_2$: doi.org/10.5067/IS-40e/TEMPO/NO2_L2.003; TROPOMI $NO_2$:

doi.org/10.5270/S5P-9bnp8q8; OCI UAA AOD on PACE: doi.org/10.5067/PACE/OCI/L2/AER_UAA/3.0; MODIS AOD on Aqua: doi.org/10.5067/MODIS/MYD04_L2.061; MODIS AOD on Terra: doi.org/10.5067/MODIS/MOD04_L2.061; VIIRS DT AOD on Suomi-NPP: doi.org/10.5067/VIIRS/AERDT_L2_VIIRS_SNPP.002; VIIRS DT AOD on NOAA-20: doi.org/10.5067/VIIRS/AERDT_L2_VIIRS_NOAA20.002; VIIRS DB AOD on Suomi-NPP: doi.org/10.5067/VIIRS/AERDB_L2_VIIRS_SNPP.002; VIIRS DB AOD on NOAA-20:

doi.org/10.5067/VIIRS/AERDB_L2_VIIRS_NOAA20.002. The doi is not available for data of AERONET, PANDORA, and AirNow, but they are publicly available. AERONET AOD data can be accessed through their API following the instructions on https://aeronet.gsfc.nasa.gov/print_web_data_help_v3_new.html. PANDORA data can be accessed from http://data.pandonia-global-network.org/. The hourly $PM_{2.5}$ and ozone data are accessible with AirNow API credentials on the webpage (https://docs.airnowapi.org/). The v5.12.4 of MERRA-2 reanalysis data is publicly available on Goddard Earth

Sciences (GES) Data and Information Services Center (DISC) (https://disc.gsfc.nasa.gov/). The doi of the meteorological condition from M2I3NVASM is 10.5067/WWQSXQ8IVFW8. The doi for aerosol mixing ratios from M2I3NVAER is 10.5067/LTVB4GPCOTK2.



**Author contribution**

SW: writing - original draft, review & editing, methodology, investigation, formal analysis, data curation; JB: writing -
original draft, review & editing, methodology; SH: writing - original draft, review & editing, conceptualization, project
administration, investigation, formal analysis, funding acquisition; CD: writing - original draft, review & editing,; BM:
writing - review & editing,; MA: writing - original draft, review & editing, data curation; CL: writing - review & editing,
conceptualization, project administration, investigation, formal analysis, funding acquisition.

**Competing interests**

The authors declare that they have no known competing financial interests or personal relationships that could have appeared
to influence the work reported in this paper.

**Acknowledgements**

We would like to acknowledge high-performance computing support from the Derecho system (doi:10.5065/qx9a-pg09) and
the Casper system (https://ncar.pub/casper) provided by the NSF National Center for Atmospheric Research (NCAR),
sponsored by the National Science Foundation. This study made use of data from NASA's Plankton, Aerosol, Cloud, ocean
Ecosystem (PACE) mission, distributed by the NASA Ocean Biology Distributed Active Archive Center (OB.DAAC). The
AOD data from MODIS C6.1 and VIIRS DT and DB were acquired from the Level-1 and Atmosphere Archive &
Distribution System (LAADS) Distributed Active Archive Center (DAAC), located in the Goddard Space Flight Center in
Greenbelt, Maryland (https://ladsweb.nascom.nasa.gov/). The data of Modern-Era Retrospective analysis for Research and
Applications, Version 2 (MERRA-2), produced by the NASA Global Modeling and Assimilation Office (GMAO) is
obtained from the NASA Goddard Earth Sciences Data and Information Services Center (GES DISC)
(https://disc.gsfc.nasa.gov/).

**Financial support**

This study was jointly supported by the NOAA Science Collaboration Program (NA21OAR4310383), through the Joint
Polar Satellite System (JPSS) Proving Ground and Risk Reduction Program, and by the NASA PACE Science and
Applications Team (Grant No. 80NSSC24K1786).



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
