# Peer review of "Leveraging JEDI for Atmospheric Composition: A unified framework for evaluating observations and model forecasts"

_EGUsphere, 2025_

## Referee Comment (RC1)

Title: Leveraging JEDI for Atmospheric Composition: A unified framework for evaluating

observations and model forecasts

Authors: Wei et al.

**General comments**

This manuscript describes leveraging JEDI components, along with MetPlus, to evaluate model

forecasts against observations for the atmospheric composition applications, mostly benefiting

from JEDI's modular and model-agnostic design. The authors demonstrated this capability

through various cases comparing model equivalents from wrf-chem and MERRA-2, with

widely-used observations in this community such as trace gases, AOD, surface PM2.5 in the

ground-based or spaceborne platforms. As pointed out by the authors, the manuscript's focus is

more on a technical demonstration of this capability rather than scientific findings. This is

appropriate and falls within the GMD scope. To possibly help its scientific aspect, however, I

have the following general comments.

1. Some of this described capability has been explored and tested in multiple JEDI-related

scientific publications. I would encourage the authors to do more relevant literature

review and to cite these papers where appropriate. It should help complement the current

technical focus on this manuscript by supplementing additional scientific validations

from others

2. Description and interpretation of some figures and tables is insufficient or irrelevant

(please see specific comments below). In several instances, the manuscript simply says

JEDI has this capability and then refers to figures or tables, with no further explanatory

description or summarization about them. While this manuscript is more

technical-focused, an accurate and even short description of included figures or tables

would be needed for a scientific publication to enhance readers' understanding.

As JEDI is undergoing rapid development and will be widely used for future research and

operations in our community, this manuscript would be a great contribution. Overall, this

manuscript is very nicely written and organized. I believe the above comments are easy to

address. Therefore, my recommendation to the editor is minor revision (see specific comments as

follows).

**Specific comments/Technical corrections**

• Line 100: Typo: "Model forecasts states" → "Model forecast states"

1

- Section 3.7: Please clarify VIIRS AOD used in this manuscript is NOAA product or NASA product. Because their algorithms are slightly different over the land.
- Figure 3: Please add some interpretation for Fig. 3 and be more relevant and accurate. Otherwise, it is not clear why it is included here. Especially, when compared with different obs, the model forecast opposite biases, e.g., slightly positive bias wrt TROPOMI and large negative bias wrt TEMPO. How will this "offers insight into systematic over- or underestimation patterns in time that can inform on atmospheric model process and emission inventory improvements"? Are comments about whether to show diurnal cycles in TEMPO and TROPOMI around Line 315 reflected in Fig. 3?
- Table 1: Similar comments as Figure 3.
- Line 360: Is VIIRS AOD NOAA/NESDIS product or the NASA product, because they apply different algorithms
- Figure 7, 8 and Table 2: Similar comments as Figure 3. It's not clear why they are suggestive of "This highlights for example the utility of AERONET as a benchmark for both model evaluation and observation operator performance.".
- Near Line 410: H(x) is called "observation equivalents", but "model equivalents" elsewhere. Please be consistent and use "model equivalents" throughout the manuscript.

---

## Author Comment (AC1)

**Response to RC #1**

Thank you very much for taking the time to review this manuscript and providing constructive comments and suggestions. Please find the detailed point-to-point responses below.

1. Some of this described capability has been explored and tested in multiple JEDI-related scientific publications. I would encourage the authors to do more relevant literature review and to cite these papers where appropriate. It should help complement the current technical focus on this manuscript by supplementing additional scientific validations from others.
*Response:* Thanks for the comment. Studies from atmospheric composition to meteorological analysis using JEDI framework have been added to Section 2.1.

2. Description and interpretation of some figures and tables is insufficient or irrelevant (please see specific comments below). In several instances, the manuscript simply says JEDI has this capability and then refers to figures or tables, with no further explanatory description or summarization about them. While this manuscript is more technical-focused, an accurate and even short description of included figures or tables would be needed for a scientific publication to enhance readers' understanding.
*Response:* We have included brief interpretations for the corresponding paragraphs of each figure in the revision.

As JEDI is undergoing rapid development and will be widely used for future research and operations in our community, this manuscript would be a great contribution. Overall, this manuscript is very nicely written and organized. I believe the above comments are easy to address. Therefore, my recommendation to the editor is minor revision (see specific comments as follows).

Specific comments/Technical corrections

- Line 100: Typo: "Model forecasts states" → "Model forecast states"
  *Response*: It has been revised.
- Section 3.7: Please clarify VIIRS AOD used in this manuscript is NOAA product or NASA product. Because their algorithms are slightly different over the land.
  *Response*: We use Dark Target and Deep Blue products developed by NASA. The paragraph has been revised for clarity.
- Figure 3: Please add some interpretation for Fig. 3 and be more relevant and accurate. Otherwise, it is not clear why it is included here. Especially, when compared with different obs, the model forecast opposite biases, e.g., slightly positive bias wrt TROPOMI and large negative bias wrt TEMPO. How will this "offers insight into systematic over- or underestimation patterns in time that can inform on atmospheric model process and emission inventory improvements" ? Are comments about whether to show diurnal cycles in TEMPO and TROPOMI around Line 315 reflected in Fig. 3?

*Response*: A brief interpretation based on the newly added CRMSE figure has been provided for the revision. See L325-332.

- Table 1: Similar comments as Figure 3.
  *Response*: See response above.
- Line 360: Is VIIRS AOD NOAA/NESDIS product or the NASA product, because they apply different algorithms
  *Response*: It is the NASA product. See the response to the comment for Section 3.7 above.
- Figure 7, 8 and Table 2: Similar comments as Figure 3. It's not clear why they are suggestive of "This highlights for example the utility of AERONET as a benchmark for both model evaluation and observation operator performance.".
  *Response*: We revisit the statement and confirm that we cannot conclude it based on the cross-comparison in Figure 7, although AERONET measurements are widely used as the benchmark to evaluate models and validate retrieval algorithms. Therefore, the sentence has been removed. Besides, we have provided more interpretations for the cross-comparison of AOD products in the revision.
- Near Line 410: H(x) is called "observation equivalents", but "model equivalents elsewhere. Please be consistent and use "model equivalents" throughout the manuscript.
  *Response*: Thanks for pointing out the inconsistencies. It has been updated to "observation equivalents" throughout the revised manuscript.

---

## Author Comment (AC2)

**Response to RC #2**

Thank you very much for taking the time to review this manuscript and providing constructive comments and suggestions. Please find the detailed point-to-point responses below.

The paper presents how the JEDI framework and associated tools can be used to evaluate the atmospheric composition models. It is interesting to see that the JEDI framework can be used beyond data assimilation. This capability of JEDI would be of interest to atmospheric chemistry researchers. It allows using JEDI in both data assimilation and verification within the same atmospheric modeling systems. The text is concise and easy to follow. I recommend publishing this manuscript after addressing the following comments:

- My main concern regarding this evaluation framework is that the authors do not show how the uncertainties are accounted for in the atmospheric composition observations. These uncertainties are taken into account in the data assimilation, but not much in model verification. For example, there are significant uncertainties in the satellite AOD observations. How can these uncertainties be taken into account in the model evaluation? This aspect of the model-observation comparison is not considered here.

  *Response:* You are correct that both observations and model forecasts have their own sources of uncertainty. Also noted that there is a large uncertainty or systematic error in the observation operator used to calculate the observation equivalent (i.e., H(x)). For instance, AOD is only a proxy for three-dimensional atmospheric constituents, and the way it is retrieved from remote sensing data is fundamentally different from how numerical models diagnose AOD as a vertically integrated quantity across aerosol species. Hence, model-observation comparisons are not only affected by observational uncertainty but also by uncertainties inherent in the H(x). Even when these uncertainties are not explicitly accounted for, however, it is still meaningful and crucial to characterize the error statistics of a given model or observation type in a consistent and systematic manner. The cross-comparison we presented in Figures 6-7 for AOD (Figure 2 for trace gas) can help identify such uncertainties. While it is not straightforward to specify all the uncertainty in the observations that vary in space and time, once they are characterized, it should be feasible to engineer them into the JEDI system. We hope this study can open the door towards more objective and consistent verification practices for both the observational and modeling communities, given that the JEDI system can provide a unified framework to investigate and cross-check uncertainties associated with observations, models, and observation operators - either individually or collectively - routinely.

- How can these tools be used to conduct object based verification?

  *Response:* This tool generates sequential observation data in IODA format. The object-based verification is not developed and can be explored in the future.

- How about aircraft observations? For example, the observations obtained during the field campaigns.

  *Response:* As long as the data is converted to the IODA format and the corresponding forward operator is available in UFO, then the data can be used to verify with model forecasts.

- 160: Does CRTM allow treatment of aerosols with non-spherical shapes?

  *Response:* With the CRTM default table, all aerosols are assumed to be spherical. With the optional GOCART-GEOS5 table, dust is assumed non-spherical following the aerosol configuration in GOCART. The information and citation have been provided in the revision.

- 230: The AirNow network is run by the states.

  *Response:* Thanks for the comment. The sentence has been revised as follows: "The AirNow system is a cooperative platform across agencies, including the U.S. Environmental Protection Agency (EPA), NOAA, National Park Service, NASA, Centers for Disease Control, and tribal, state, and local air quality agencies." (L240)

- 275: I assume the WRF-Chem simulations used the biomass burning emissions generated sometime later. Therefore, I would not say these are "WRF-Chem forecasts," but rather retrospective simulations. Add a Table to show the model settings, e.g. resolution, physics and chemistry schemes that were used in the WRF-Chem simulations.

  *Response:* To the best of our knowledge, the terms "forecasts" and "simulations" are used interchangeably, even for retrospective case studies. The WRF-Chem simulation in this work uses the near-real-time FINN emissions generated in the previous day to conduct 54-hour forecasts at 00Z every day. However, in response to your comment, the description of the model has been updated to the revised manuscript (L288) to provide information on the near-real-time forecast from the system. Regarding your request to add a new table, we have chosen to keep the focus of this paper on the evaluation capabilities provided by the JEDI system, rather than on the details of our specific model setup. We therefore limit the description to introducing WRF-Chem as one of the models interfaced with JEDI. Thank you for your understanding.

- It is remarkable that the WRF-Chem model has strong underestimation of the NO2 columns over NYC. What is causing this? The emission inventories or some other model errors. How does this bias affect O3 simulations over NYC?

  *Response:* We agree that the evaluation capability demonstrated in this study can help uncover many interesting features and, in turn, motivate further scientific investigations. But these model-specific questions go beyond the scope of this study. As such, we leave them for future studies.

- The RMSE statistics: Is this the centered RMSE? If not, I suggest changing that as the bias statistics are already provided.

  *Response:* Thanks for the comment. The CRMSE has been provided in the revision.

- Figure 5. This map does not provide much information. It would be better to replace it with two plots to show the PM2.5 and O3 bias maps.

  *Response:* The bias maps of PM2.5 and O3 have been provided in the revision.

- MERRA-2 evaluation: As it is stated, MERRA-2 is based on GEOS-5, which assimilates the AERONET observations. Therefore, it is not clear what is the purpose of the MERRA-2 evaluation using the AERONET data. The presented AOD satellite verification is quite useful.

  *Response:* As Randles et al. (2017) reported, the assimilation of AERONET data is up to October 2014 because the data is not available in near-real-time. More descriptive information has been provided in the revised paragraphs.

- Figure 7: Please add the regression lines (slope and intercept)

  *Response:* The regression line and the corresponding slope and intercept have been provided in the updated Figure 7.

- References: For WRF/WRF-Chem this paper should be cited too: Powers, J. G., Klemp, J. B., Skamarock, W. C., Davis, C. A., Dudhia, J., Gill, D. O., et al. (2017). THE WEATHER RESEARCH AND FORECASTING MODEL Overview, System Efforts, and Future Directions. *Bulletin of the American Meteorological Society*, *98*(8), 1717–1737

  *Response:* Thanks for the suggestion. The reference has been added to the revision.